# A deletion polymorphism in the *Caenorhabditis elegans* RIG-I homolog disables viral RNA dicing and antiviral immunity

**Alyson Ashe[1,2†], Tony Bélicard[3,4†], Jérémie Le Pen[1,2†], Peter Sarkies[1,2†], Lise Frézal[3,4], Nicolas J Lehrbach[1,2‡], Marie-Anne Félix[3,4*], Eric A Miska[1,2*]**

[1]Wellcome Trust/Cancer Research UK Gurdon Institute, University of Cambridge, Cambridge, United Kingdom; [2]Department of Biochemistry, University of Cambridge, Cambridge, United Kingdom; [3]Institute of Biology of Ecole Normale Supérieure (IBENS), Centre National de la Recherche Scientifique, UMR 8197, Paris, France; [4]Institut National de la Santé et de la Recherche Médicale U 1024, Paris, France

**Abstract** RNA interference defends against viral infection in plant and animal cells. The nematode *Caenorhabditis elegans* and its natural pathogen, the positive-strand RNA virus Orsay, have recently emerged as a new animal model of host-virus interaction. Using a genome-wide association study in *C. elegans* wild populations and quantitative trait locus mapping, we identify a 159 base-pair deletion in the conserved *drh-1* gene (encoding a RIG-I-like helicase) as a major determinant of viral sensitivity. We show that DRH-1 is required for the initiation of an antiviral RNAi pathway and the generation of virus-derived siRNAs (viRNAs). In mammals, RIG-I-domain containing proteins trigger an interferon-based innate immunity pathway in response to RNA virus infection. Our work in *C. elegans* demonstrates that the RIG-I domain has an ancient role in viral recognition. We propose that RIG-I acts as modular viral recognition factor that couples viral recognition to different effector pathways including RNAi and interferon responses.

*For correspondence: felix@
biologie.ens.fr (M-AF); eric.
miska@gurdon.cam.ac.uk (EAM)

†These authors contributed
equally to this work

‡Present address: Department
of Molecular Biology, Harvard
Medical School, Boston, United
States

**Reviewing editor**: Detlef Weigel,
Max Planck Institute for
Developmental Biology,
Germany

## Introduction

The arms races between pathogens and their hosts have led to the evolution of sophisticated mechanisms to provide immunity against infection. Whilst adaptive immunity is specific to vertebrates, innate mechanisms are present in all multicellular organisms, allowing cells to recognize specific pathogens and instigate appropriate responses. RNA viruses are important pathogens of many multicellular organisms, which replicate without a DNA intermediate using RNA dependent RNA polymerase. Successful neutralization of invading RNA viruses by cells thus requires the viral genome to be recognized within the sea of endogenous RNA.

The primary innate immune sensors for RNA viruses in mammals are RIG-I and its homolog MDA-5 (*Schlee, 2013*). Viral recognition by RIG-I and MDA-5 triggers activation of downstream signaling, mediated by the proteins' N-terminal CARD domains, and results in the activation of the interferon pathway (*Yoneyama et al., 2004*). Initial recognition of viral RNA is likely to be mediated by the DExD/H-box helicase domain and the C-terminal RIG-I domain. Though the precise ligands that activate RIG-I family proteins are not fully defined, MDA-5 appears to bind long dsRNA (*Gitlin et al., 2006*), whilst RIG-I seems to recognize the 5′ end of double stranded RNA, but only if it has a 5′ triphosphate (*Pichlmair et al., 2006*). As all known RNA polymerases leave a triphosphate at the 5′ end of newly synthesized RNA, the presence of a 5′ triphosphate is likely to be a signature of RNA virus replication

**eLife digest** Most organisms—from bacteria to mammals—have at least a rudimentary immune system that can detect and defend against pathogens, particularly viruses. This defense mechanism, which is known as the innate immune system, uses sensor proteins to recognize viral RNA, and then mobilizes other immune components to attack the invaders.

The specific mechanisms used to destroy viruses differ between species. In mammals, a protein called RIG-1 binds to viral RNA and activates a signaling pathway that leads to the production of interferons: immune proteins named after their ability to 'interfere' with viral replication. Plants and insects do not use interferons, but instead use a mechanism called RNA interference, in which long double-stranded RNAs are cleaved into shorter fragments.

The nematode worm *C. elegans* also deploys RNA interference against viruses but, in contrast to insects and plants, worms do not possess a specific set of RNA interference enzymes that participate solely in the antiviral response. They do, however, express a protein called DRH-1 that is related to the RIG-I protein found in mammals.

To investigate whether DRH-1 contributes to innate immunity in *C. elegans*, Ashe et al. infected 97 strains of *C. elegans* from around the world with a virus, and showed that some strains were more sensitive to the virus than others, with certain strains showing complete resistance. By comparing a sensitive strain with a resistant one, Ashe et al. revealed that viral sensitivity was caused by a mutation in the gene encoding DRH-1.

Further experiments showed that DRH-1 is required for the first step in RNA interference. Ashe et al. have thus identified a conserved role for RIG-1 in initiating antiviral responses, and propose that the protein couples virus recognition to distinct defense mechanisms in different evolutionary groups.

and thus allow viral replication intermediates to be distinguished from endogenous mRNA, which will predominantly display a 5′ cap (*Rehwinkel and Reis e Sousa, 2010*).

In plants and insects, interferon signaling is not involved in antiviral defense. Instead, the RNA interference (RNAi) pathway provides robust defense against RNA viral infection (*Ding and Voinnet, 2007*; *Ding, 2010*). The initial step in protection against positive strand RNA virus infection in plants and insects is detection and subsequent cleavage of the double-stranded replication intermediate by members of the Dicer family of endonucleases. Insects and plants possess dedicated Dicer enzymes responsible specifically for this antiviral response, Dicer-like 4 (and to a lesser extent Dicer-like 1) in plants, and Dicer2 in insects (*Bouché et al., 2006*; *Deleris et al., 2006*; *Fusaro et al., 2006*; *Galiana-Arnoux et al., 2006*; *van Rij et al., 2006*; *Diaz-Pendon et al., 2007*). The small RNAs thus generated feed into the canonical RNAi machinery and can be used to silence the viral genome.

In the nematode *C. elegans*, the RNAi pathway is best characterized as a response to artificial intro-duction of dsRNA by feeding or injection. However, exposing *C. elegans* to virus-derived dsRNA using transgenes or infection with mammalian viruses also triggers an RNAi response (*Lu et al., 2005*; *Schott et al., 2005*; *Wilkins et al., 2005*; *Yigit et al., 2006*). Additionally, we have shown previously that this response is also initiated upon infection with a positive strand RNA virus, named the Orsay virus, which infects *C. elegans* in the wild through horizontal transmission. Disruption of this pathway through mutation of the core components of the RNAi machinery results in greatly increased viral infection levels. Interestingly, the Orsay virus infects a wild isolate named JU1580 (from which the Orsay virus was isolated) to much higher levels than the N2 strain (*Félix et al., 2011*).

Despite these advances, our understanding of how RNAi-mediated antiviral defense in *C. elegans* is orchestrated is still limited. One major unsolved problem is how viral dsRNA within the cell is recognized in order to initiate the antiviral response. In contrast to the situation in both plants and insects, *C. elegans* only has one Dicer enzyme (*Knight and Bass, 2001*; *Yigit et al., 2006*); it is expected therefore that the activity of Dicer in different small RNA pathways will be controlled by different partner proteins within distinct complexes (*Yigit et al., 2006*; *Thivierge et al., 2012*).

In this regard it is intriguing that *C. elegans* encodes three homologues of the mammalian viral recognition protein RIG-I: DRH-1, DRH-2 and DRH-3. The three RIG-I family members do not contain the CARD domains required for interferon induction; consistently they have yet to be implicated in

signaling pathways. However, they do possess the RIG-I C-terminal domain and the helicase domains, potentially enabling them to recognize viral RNA. So far study of these genes has connected them to the RNA interference response. DRH-1 was initially characterized as a protein interacting with DCR-1, RDE-1 and RDE-4 (*Tabara et al., 2002*; *Duchaine et al., 2006*; *Sijen et al., 2007*). RNAi of *drh-1* was not found to be required for RNAi per se but result in a defect in RNAi of a second gene (*Tabara et al., 2002*), however it remains unclear whether this effect was solely due to knockdown of *drh-1*. More recent studies did not observe defects in endogenous or exogenous small RNA pathways in *drh-1* mutants (*Gu et al., 2009*; *Lu et al., 2009*). However, *drh-1* mutants were found to be defective in the silencing of a flockhouse virus-derived replicon (*Lu et al., 2009*). *drh-1* and *drh-2* are the result of a recent gene duplication event that occurred after the last common ancestor of *C. elegans* and its closest known sister clade including *C. briggsae* (www.wormbase.org) (*Stein et al., 2003*), and *drh-2* is the upstream gene in an operon containing *drh-1*. *drh-2* has lost some of its functional domains due to frame-shift mutations and its function remains unclear, although it has been suggested to act as a negative regulator of RNAi (*Lu et al., 2009*). DRH-3 is required for endogenous small RNA pathways in the germline and efficient exogenous RNAi (*Gu et al., 2009*), and is found in at least two distinct protein complexes including the ERI Complex (ERIC) (*Gu et al., 2009*; *Thivierge et al., 2012*). Thus, an intriguing possibility is that RIG-I family genes in *C. elegans* initiate RNAi rather than the interferon response.

Here we discover a naturally occurring deletion in the gene *drh-1* that is widespread in *C. elegans* populations despite predisposing individuals to viral sensitivity. We show that the increased viral sensitivity caused by this deletion results from a failure of the RNAi pathway. Mutations in *drh-1* almost completely abolish the production of primary siRNAs, allowing us to place DRH-1 at the top of a hierarchical RNAi response. Our data supports a model whereby the conserved RNA virus recognition capability of *drh-1* allows it to recruit the RNAi machinery to defend *C. elegans* from viral infection.

## Results

### A natural *drh-1* deletion is a major determinant of viral sensitivity in *C. elegans*

To explore intraspecific variation in viral resistance in *C. elegans*, we assayed a worldwide set of 97 wild *C. elegans* isolates that had previously been genotyped (*Andersen et al., 2012*). To assess viral sensitivity, we infected each isolate in triplicate and quantified the viral load after 7 days by qRT-PCR. Viral loads of the 97 isolates varied widely over five orders of magnitude (*Figure 1—figure supplement 1A*). Genome-wide association for viral load revealed a single peak covering a 6 Mb region in the middle of chromosome IV (*Figure 1A*). However, further mapping resolution is limited by the low natural recombination frequency in the species (*Cutter et al., 2009*; *Andersen et al., 2012*).

We therefore focused on genetic variation in viral sensitivity between the N2 and JU1580 isolates. We first assayed viral sensitivity in 110 F2 recombinant families (*Figure 1—figure supplement 1B*). Whole-genome sequencing of a pool of the 20 most sensitive families revealed linkage to chromosome IV (*Figure 1—figure supplement 1C*). To narrow down the candidate region, we introgressed the center of chromosome IV (IV:3,329,219 to IV:11,083,410) of JU1580 into N2 animals (yielding strain JU2170). As expected, JU2170 showed similar viral load to JU1580 (*Figure 1—figure supplement 2*). We then screened for recombinants in this region after crossing JU2170 with N2. This allowed us to restrict the candidate region to a 155 kb interval carried by a virus-sensitive recombinant strain named JU2196 (*Figure 1B*, *Figure 1—figure supplement 2*).

Genome sequencing of the JU1580 isolate and alignment to the N2 reference strain revealed 20 single nucleotide polymorphisms (SNPs), including one non-synonymous SNP, and one 159 base indel in the 155 kb region (*Figure 1—figure supplement 2*). The 159 base deletion in the JU1580 genome, named *niDf250* (IV:6,607,635-6,607,793), lies within the *drh-1* gene locus (*Figure 1C*). *drh-1* is a homolog of the mammalian *RIG-I* family genes, whose products bind virus-derived RNAs, acting as pattern recognition receptors (*Parameswaran et al., 2010*; *Yoneyama and Fujita, 2010*), and trigger antiviral innate immune responses in mammals (*Yoneyama et al., 2004*). *C. elegans* expresses three RIG-I-like proteins: DRH-1, DRH-2 and DRH-3. The *niDf250* deletion covers most of *drh-1* exon 19 and part of exon 20 (*Figure 1C*) and was confirmed using genomic PCR (*Figure 1D*). *drh-1* mRNA levels are unchanged in JU1580 (*Figure 1E*) and the resulting transcript is predicted to encode a truncated protein of 987 amino acids, identical to the N2 form of DRH-1 up to amino acid 973 but with

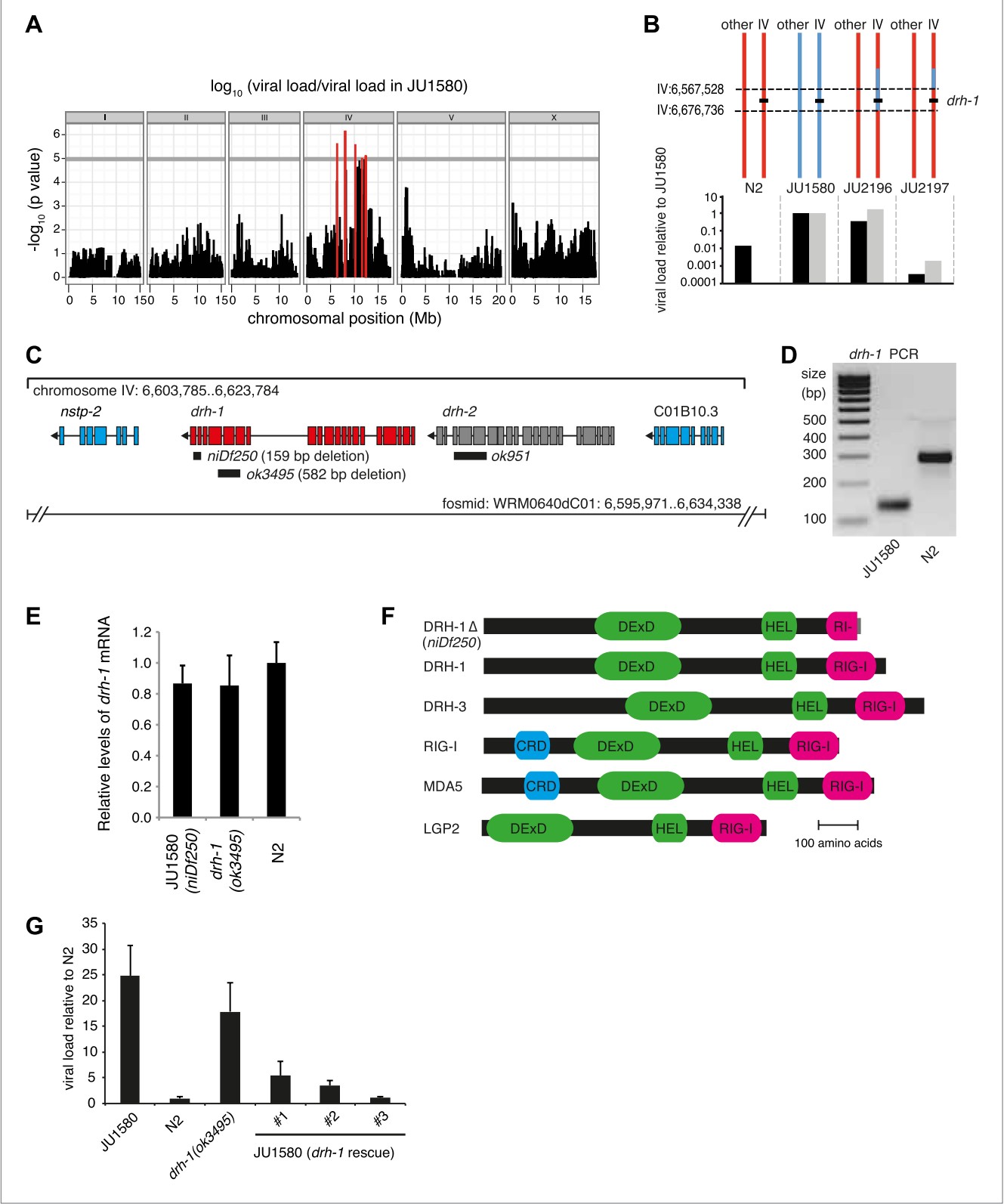

**Figure 1**. A deletion polymorphism in *drh-1* is a major determinant of Orsay virus sensitivity in wild isolates of *C. elegans*. (**A**) Genome-wide association analysis of Orsay virus sensitivity in 97 wild isolates of *C. elegans*. The mapped trait is the viral load of animals, measured by qRT-PCR on the Orsay virus RNA2 genome after 7 days of infection, using three independent infection experiments. The horizontal grey line is a Bonferroni-corrected threshold of
*Figure 1. Continued on next page*

*Figure 1. Continued*

significance at p=0.05. Peaks reaching above this threshold are colored in red. (**B**) Fine mapping of the candidate region causative for virus hypersensitivity observed in JU1580 animals. The genotypes of chromosome IV and other chromosomes are represented for parental (N2 and JU1580) and informative recombinant (JU2196 and JU2197) strains. Regions in red or blue are identical to N2 or JU1580, respectively. The inferred candidate region is delimited by dotted lines. Below each genotype are viral load measured by qRT-PCR of Orsay virus RNA2, in two independent infections (black and grey bars) and normalized to JU1580. (**C**) Diagram of the *drh-1* locus. Positions of deletion alleles and a rescuing fosmid are indicated. (**D**) PCR analysis of *niDf250* deletion in N2 and JU1580 strains. (**E**) *drh-1* mRNA level in different strains (as indicated), measured by RT-qPCR. (**F**) Diagram of *C. elegans* and human RIG-I like genes. DeXD = Pfam:DEAD, Hel = Pfam:Helicase_C, RIG-I = Pfam:RIG-I_C-RD, CRD = Pfam:CARD. (**G**) Viral load in different strains (as indicated), measured by RT-qPCR of the Orsay virus RNA1 genome after 4 days of infection. JU1580 (*drh-1* rescue) refers to JU1580 strains carrying three independent transgenic lines (SX2375, SX2376, SX2377). Transgenes include the fosmid WRM0640dC01 and a co-injection marker, they were integrated into the genome using X-rays. Error bars represent the standard error of the mean (SEM) of five biological replicates.

The following figure supplements are available for figure 1:

**Figure supplement 1**. Variation in the ability of the Orsay virus to replicate in *C. elegans*.

**Figure supplement 2**. Genotype and sensitivity to the Orsay virus of recombinants in the chromosome IV region.

a novel C-terminus (*Figure 1F*). Importantly, this truncates the RIG-I C-terminal domain (amino acids 885–1014), thought to be required for RNA recognition specificity (*Kowalinski et al., 2011*; *Figure 1F*).

To establish a potential role for DRH-1 in Orsay antiviral resistance we tested whether a *drh-1* deletion in the N2 background was sufficient to impart viral sensitivity, as assayed by viral load. Indeed, the *drh-1(ok3495)* mutant displayed an increased viral load compared to N2 animals, similar to that of JU1580 (*Figure 1G*). Conversely, transgenic JU1580 animals carrying a fosmid containing the N2 allele of *drh-1* were resistant to Orsay infection (*Figure 1G*). Therefore, variation at the *drh-1* locus explains the difference in viral load between N2 and JU1580.

An inactivating mutation in a pathogen-resistance gene could be expected to be a rare deleterious variant in natural populations, yet the wild *niDf250* allele is found at an intermediate frequency at the global level, in 22/97 (23%) of the tested wild isolates. The deletion is found in about one third of isolates from Europe and Africa (21/64, 33%) with a high incidence in France (14/30, 47%), but is rarer (1/30) in those from the Americas and the Pacific regions (*Figure 2A*). As expected, the presence of the deletion correlated strongly with viral load in the infection experiment (*Figure 1—figure supplement 1A*; Wilcoxon test on viral load of isolates carrying each *drh-1* allele, p=1.3×10$^{-9}$). Thus, surprisingly, the derived *drh-1* allele has spread to intermediate frequency in natural populations, despite rendering the animals susceptible to viral infection.

A possible interpretation for the spread of the sensitive *drh-1* allele might be that high Orsay viral load has no deleterious effect on fitness. However, in laboratory conditions, we found that viral infection leads to delayed and decreased total progeny of JU1580 and *drh-1(ok3495)* mutant animals relative to uninfected animals, whilst infection of N2 had no significant effect (*Figure 2—figure supplement 1*). Furthermore, we performed a competition experiment between N2 and JU2196, which contains the *drh-1* region from JU1580 introgressed into the N2 background (*Figure 1B*). N2 rapidly outcompetes JU2196 in the presence of viral infection, but not its absence, confirming that the increased viral infection resulting from the *drh-1* deletion is indeed detrimental for fitness (*Figure 2B*). In the absence of viral infection, we could not detect in standard laboratory conditions over 10 generations of competition any positive or negative effect of the *drh-1* deletion and the introgressed surrounding region. Thus the natural *drh-1* deletion impairs fitness only in the presence of viral infection.

To characterize the evolutionary history of the *drh-1* region, we first focused on the 6 Mb region detected by genome-wide association. This region presents three main haplotypes among the 97 wild isolates (*Andersen et al., 2012*): the N2 haplotype, the JU1580 haplotype and a distant haplotypic group (including RW7000 and QX1211), as well as a few recombinants (*Figure 2C*, *Figure 2—figure supplement 2*). The *drh-1(niDf250)* allele is exclusively found in isolates carrying the JU1580 haplotype in the 6 Mb region and in a few recombinants with either the N2 or distant haplotypes (*Figure 2C*). The JU1580 and N2 haplotypes show fewer fixed differences in RAD polymorphic sites between them (24 SNPs) than with the divergent haplotype group (63 and 57 SNPs, respectively). In addition, from our whole-genome JU1580 sequencing data, N2 and JU1580 display a very low level of molecular diversity

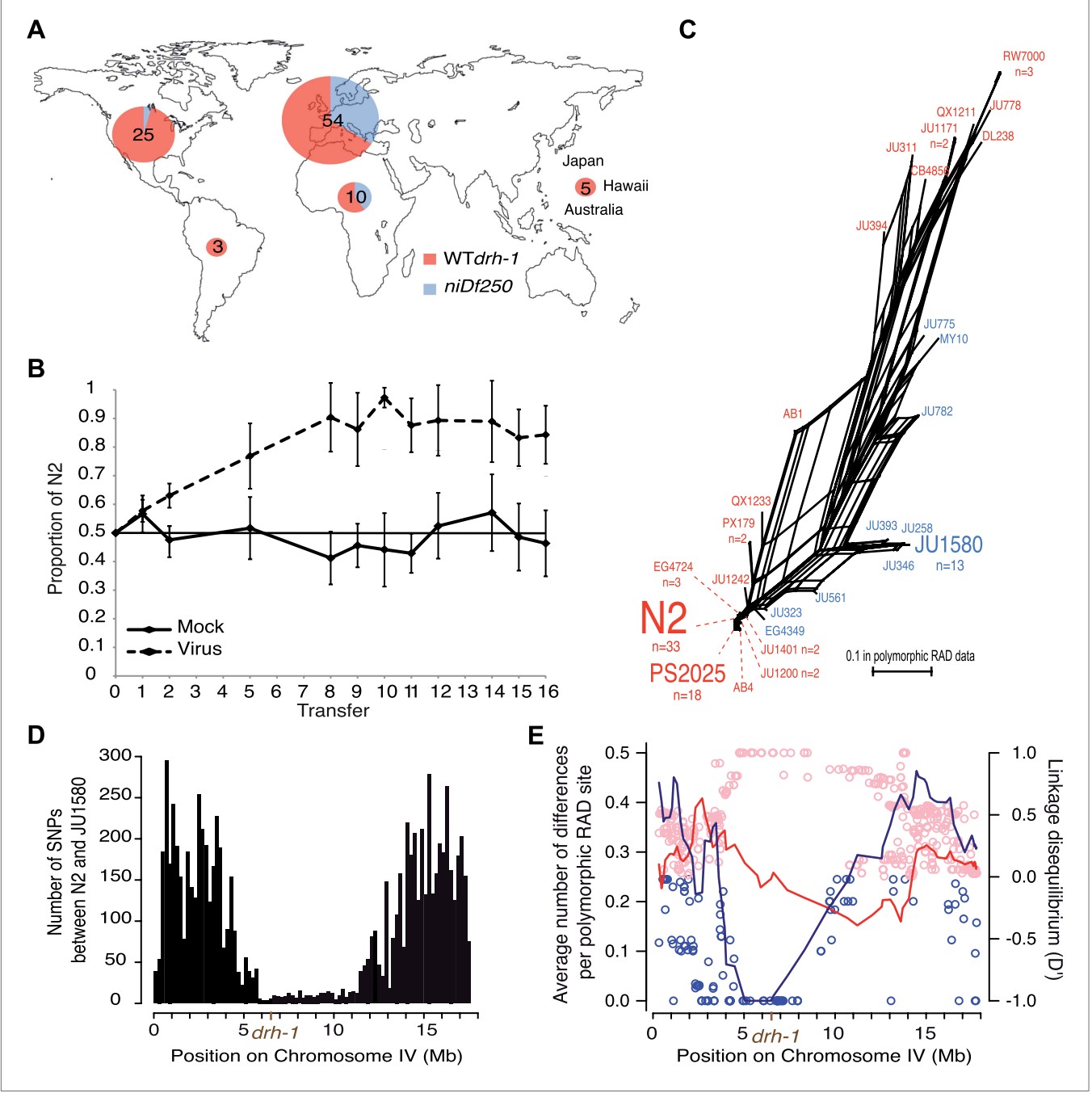

**Figure 2**. Geographic distribution and evolutionary genetic context of *drh-1* alleles. (**A**) Geographic distribution of *drh-1* alleles. The respective frequencies of the *niDf250* and N2 alleles of *drh-1* are represented for each world region in blue and red, respectively, based on genotyping of the 97 wild isolates. (**B**) Competition experiment between the N2 reference and the JU2196 introgression line. In the absence of the Orsay virus, the proportion of the N2 genotype remains close to 50% throughout the experiment (48.6 ± 5.3%). In the presence of the Orsay virus, the proportion of the N2 genotype increases and appears to stabilize around 90% after nine transfers (88.1 ± 4.6%). The presence of the virus has a significant effect (linear model, p=1.6 × 10$^{-4}$). Error bars represents standard deviation. (**C**) Neighbor-network of the 97 isolates in the chromosome IV central region associated with Orsay virus sensitivity. Only one isolate per haplotype is represented; font size is relative to the number (n) of isolates sharing this haplotype. Haplotypes in blue or red carry the *niDf250* allele of *drh-1*, respectively. (**D**) Distribution of SNPs along chromosome IV between N2 and JU1580, based on JU1580 whole-genome sequencing. (**E**) Molecular diversity (left y axis scale) is plotted along chromosome IV for isolates carrying the *niDf250* or the N2 allele as blue or red lines, respectively. Linkage Disequilibrium *D'* values (right y axis scale) between polymorphic RAD sites along chromosome IV and the *niDf250* or N2 alleles of *drh-1* are represented with blue or red circles, respectively.

*Figure 2. Continued on next page*

*Figure 2. Continued*

The following figure supplements are available for figure 2:

**Figure supplement 1**. Infection by the Orsay virus has an effect on progeny production of *drh-1* deleted strains and on longevity of the *drh-1(ok3495)* mutant.

**Figure supplement 2**. Chromosome IV haplotypes for the 97 isolates (modified from Supplemental Figure 7 in *Andersen et al., 2012*).

in the central region of chromosome IV (*Figure 2D*). Moreover, we observe a strong decrease in molecular diversity between IV:4529464 to IV:6662701 in isolates carrying the *niDf250* allele (*Figure 2E*, lines), but not in those carrying the *drh-1* (N2) allele. Thus, the divergence between the N2 and JU1580 haplotypes in this region, including the *niDf250* deletion, appears recent relative to much of the species' genetic diversity. Furthermore, the *niDf250* allele is in high or even full linkage disequilibrium with a large region of chromosome IV (*Figure 2E*, dots). This lack of diversity and the high linkage disequilibrium around *niDf250* suggest a partial sweep of the haplotype linked to *niDf250*. They also imply that the sensitive *drh-1* allele may have spread by hitch-hiking with a favorable allele.

## DRH-1 initiates an antiviral small RNA response

Having established the major role of *drh-1* allelic variation in natural variation of antiviral defense in *C. elegans*, we wished to understand the molecular mechanisms of DRH-1 action in the antiviral response. We previously showed that disruption of small RNA pathway genes such as *rde-1*, which encodes an Argonaute protein essential for RNAi in response to exogenous dsRNA (RDE-1), renders N2 animals as sensitive to the Orsay virus as JU1580 (*Félix et al., 2011*). DRH-1 interacts with the double-stranded RNA (dsRNA) binding protein RDE-4 and the dsRNA-specific endonuclease Dicer (DCR-1), both of which act upstream of RDE-1 in exogenous RNAi (*Tabara et al., 1999*; *Parrish and Fire, 2001*; *Barber et al., 2010*) However, DRH-1 is dispensable for exogenous RNAi (*Gu et al., 2009*). We therefore wondered if DRH-1 could act specifically to promote DCR-1 processing of long dsRNA that is produced during viral RNA replication. Indeed, we find that DCR-1 is required for viral resistance in the N2 strain (*Figure 3A*). We therefore postulated that the antiviral response involves siRNAs processed by DCR-1, which may initiate a cascade of events analogous to canonical *C. elegans* RNAi.

In *C. elegans*, the RNAi pathway is divided into primary and secondary steps. Long dsRNA is processed by DCR-1 to generate a primary siRNA duplex ~23 nucleotides (nt) in length with 5′ monophosphates and 2 nt 3′ overhangs. By examining libraries derived solely from small RNAs with 5′ monophosphates (5′ dependent libraries), we could interrogate the primary siRNA response specifically. In wild-type N2 animals, infection with virus leads to generation of predominantly 23 nt small RNAs with no first nucleotide bias, mapping both sense and antisense to the viral genome in equal proportions (*Figure 3B*). These small RNAs are likely primary DCR-1 products generated from the double-stranded intermediate of viral replication, consistent with the small RNA response against single-strand RNA viruses in insects (*Flynt et al., 2009*). Furthermore, overlapping RNAs mapping to complementary strands of the viral genome showed the 2 nt 3′ overhang characteristic of Dicer products ($p<10^{-9}$, $\chi^2$ test against a uniform distribution of overhang length) (*Figure 3C*). The same viRNA pattern was observed in *rde-1* mutants lacking the primary siRNA Argonaute protein RDE-1 (*Figure 3D*). Thus *rde-1* mutants are proficient in primary viRNA formation despite being sensitive to viral infection (*Figure 3—figure supplement 1*) (*Félix et al., 2011*). In contrast, JU1580 animals showed a markedly different pattern (*Figure 3E*). First, a much higher proportion of small RNAs derived from the sense strand than the antisense strand (88% vs 49% in N2). Second, the size distribution of viRNAs was flattened, with a greatly reduced proportion of 23 nt RNAs and an increased proportion of 15–20 nt RNAs. These shorter viRNAs were not present in libraries prepared from uninfected JU1580 or N2 controls (*Figure 3—figure supplement 2A*) and appeared to be derived from regions of strong secondary structure within the virus (*Figure 3—figure supplement 3*). The *drh-1* mutant (in an N2 strain background) displayed an identical viRNA pattern to JU1580 (*Figure 3F*). Thus JU1580 and *drh-1* mutant strains are deficient in primary siRNA generation against the virus.

Primary siRNAs act upstream of an amplification step by triggering the synthesis of secondary siRNAs antisense to targeted RNAs (*Sijen et al., 2001*). Secondary siRNAs are synthesized by RNA-dependent RNA polymerases (RdRPs), and bind to a number of secondary siRNA-specific Argonaute

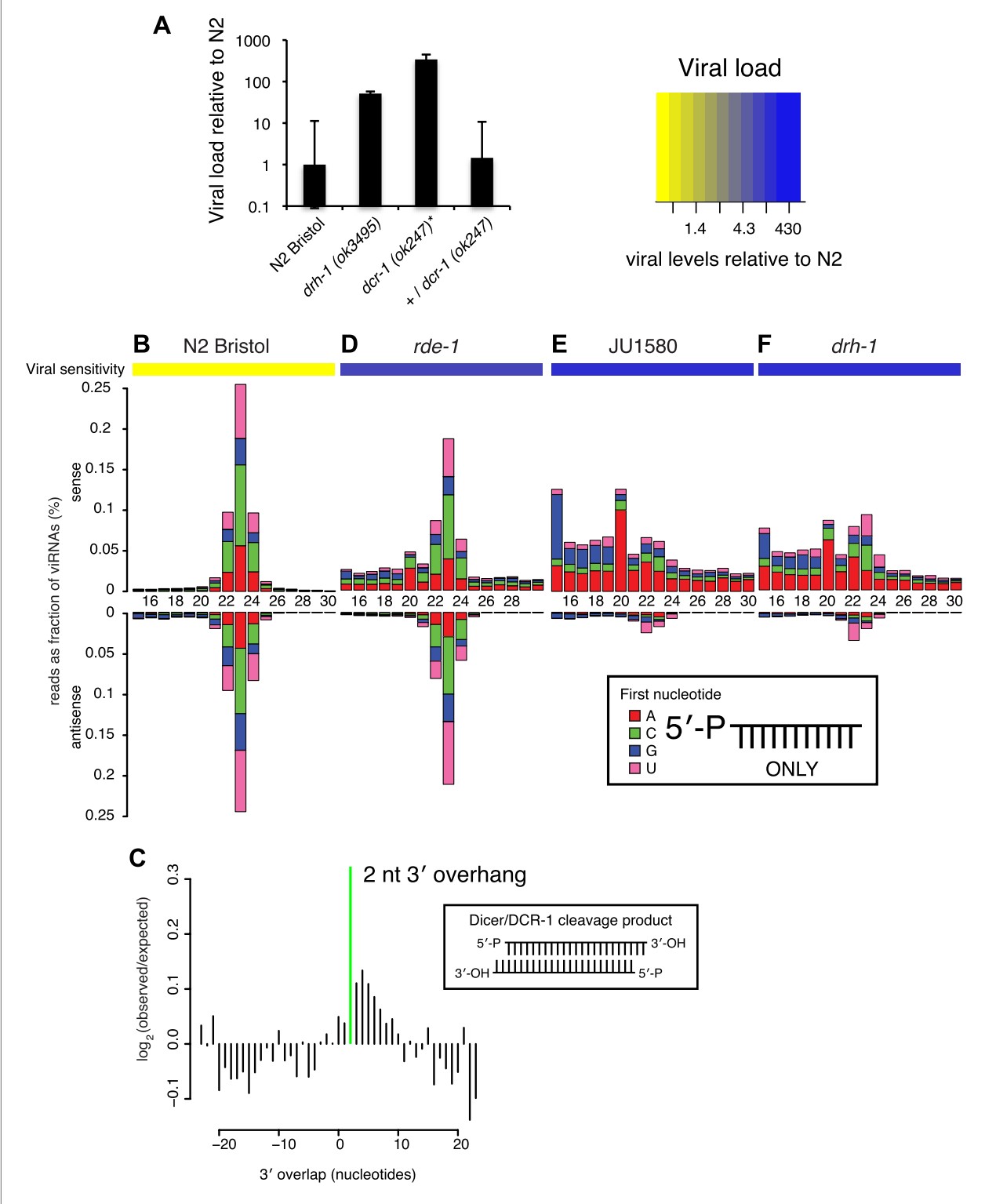

Figure 3. DRH-1 is required for the Orsay antiviral response and primary viRNA generation. (**A**) qRT-PCR analysis of viral load after 4 days of infection with the Orsay virus. *, *dcr-1* mutants are sterile, data shown are homozygous mutant animals from heterozygous mothers. (**B**) Primary viRNA populations in strains as indicated. 5' dependent small RNA sequencing captures only primary siRNAs with a 5' monophosphate. Data are grouped as sense or antisense and according to length and the identity of the first nucleotide. From the same samples viral load was measured by qRT-PCR of the Orsay virus RNA1 genome after four days of infection (heatmap, see also *Figure 3A* and *Figure 3—figure supplement 1B*). (**C**) Analysis of phasing of 23 nt primary
*Figure 3. Continued on next page*

*Figure 3. Continued*

viRNAs generated in infected N2 animals. The x axis shows the length of the overhang in nucleotides, either 5′ (negative numbers) or 3′ (positive numbers), for each pair of sequences that map to overlapping regions on opposite strands. A value of 0 represents a pair of viRNAs with perfect complementarity that would form blunt ends. The y axis shows the number of times each particular overhang was observed relative to the number of times that such an overhang would be expected if overhangs were random. Green bar indicates the 2 nt 3′ overhang. (**D–F**) same as in (**B**).

The following figure supplements are available for figure 3:

**Figure supplement 1**. Viral sensitivity in a number of mutants of small RNA pathway genes.

**Figure supplement 2**. Additional small RNA sequencing controls.

**Figure supplement 3**. Relationship between antiviral small RNAs and predicted secondary structure within the viral genome.

proteins to bring about target silencing (*Yigit et al., 2006*). Secondary siRNAs have a modal length of 22 nt, are 5′ triphosphorylated, have a strong preference for a 5′ guanine (G), and are referred to as 22G siRNAs (22Gs). Small RNAs can be enzymatically treated prior to adaptor ligation to allow sequencing (5′ independent) of both primary and secondary siRNAs (*Pak and Fire, 2007*; *Sijen et al., 2007*). N2 animals infected with Orsay virus showed a robust secondary 22G siRNA response, primarily antisense to the viral genome (*Figure 4A*, *Figure 4—figure supplement 1*). In contrast, *rde-1* mutants lacked 22G siRNAs, consistent with a role for RDE-1 in initiating the secondary siRNA response (*Figure 4B*, *Figure 4—figure supplement 1*). In addition, mutants lacking DRH-3 or the RdRP RRF-1 and a strain deficient in 12 worm Argonaute proteins (WAGO-1 through 12) that bind secondary siRNAs (MAGO-12) (*Yigit et al., 2006*) lacked 22G viRNAs, but still produced primary siRNAs (*Figure 4F,L,M*). The above have previously been implicated in secondary siRNA generation in other contexts (*Yigit et al., 2006*; *Pak and Fire, 2007*; *Sijen et al., 2007*) and *drh-3, rrf-1* and MAGO-12 mutant strains are also sensitive to viral infection similar to *rde-1* (*Figure 3—figure supplement 1*). Taken together these data show that a canonical secondary siRNA pathway is engaged to amplify the antiviral response.

*drh-1* mutants and JU1580 animals displayed the same profile of sense siRNAs in 5′ independent libraries as in 5′ dependent libraries (*Figures 3E–F and 4C,E*). Mutants deficient in the adjacent gene *drh-2* have a viRNA profile similar to N2 (*Figure 4I*) and are as resistant to viral infection as N2 (*Figure 3—figure supplement 1*), implying that this gene is not involved in the antiviral siRNA pathway. Moreover transgenic JU1580 animals carrying the *drh-1* gene from N2 showed the same overall viRNA profile as N2 (*Figure 4D*, *Figure 4—figure supplement 1*), further supporting the conclusion that *drh-1* deficiency is primarily responsible for the defective siRNA synthesis and virus sensitivity in JU1580. Importantly however, there were residual antisense 22G siRNAs present in both JU1580 and *drh-1* mutants. This suggests that the few DCR-1 products with the correct length in *drh-1* mutants can still be used to generate secondary siRNAs as in N2, implying that *drh-1* is not essential for the secondary siRNA pathway. Consistent with this interpretation, the reduced 23 nucleotide long primary siRNA products in JU1580 still displayed a 2 nucleotide 3′ overhang characteristic of DCR-1 activity (*Figure 4—figure supplement 2*).

Furthermore, residual 22G siRNAs are also present in mutants deficient in the DCR-1 accessory factor RDE-4, which is required for efficient DCR-1 activity in the exogenous RNAi pathway (*Figure 4J*; *Tabara et al., 1999*; *Parrish and Fire, 2001*). Additionally, *drh-1* mutants displayed no difference in endogenous 22G siRNAs mapping antisense to protein-coding genes, whilst, in agreement with previous data (*Gu et al., 2009*), *drh-3* mutants showed markedly reduced levels of endogenous siRNAs (*Figure 4—figure supplement 3*). Together with our observation that *drh-1* mutants are deficient in primary viRNA production, this suggests that DRH-1 acts early in the antiviral siRNA pathway and is not required for downstream amplification steps. These observations are in contrast to earlier work describing a role for DRH-1 downstream of secondary siRNA production in a flockhouse virus replicon model (*Lu et al., 2009*).

To further test this interpretation, we examined double mutant strains. The prominent 23 nt peak for sense and antisense viRNAs attributed to DCR-1 activity present in *drh-3* and *rde-1* single mutants was absent in both *drh-3; drh-1* and *drh-1; rde-1* double mutants (*Figure 4G,H*). This is consistent with the idea that DRH-1 acts upstream of DRH-3 and RDE-1 in the antiviral siRNA pathway and acts in

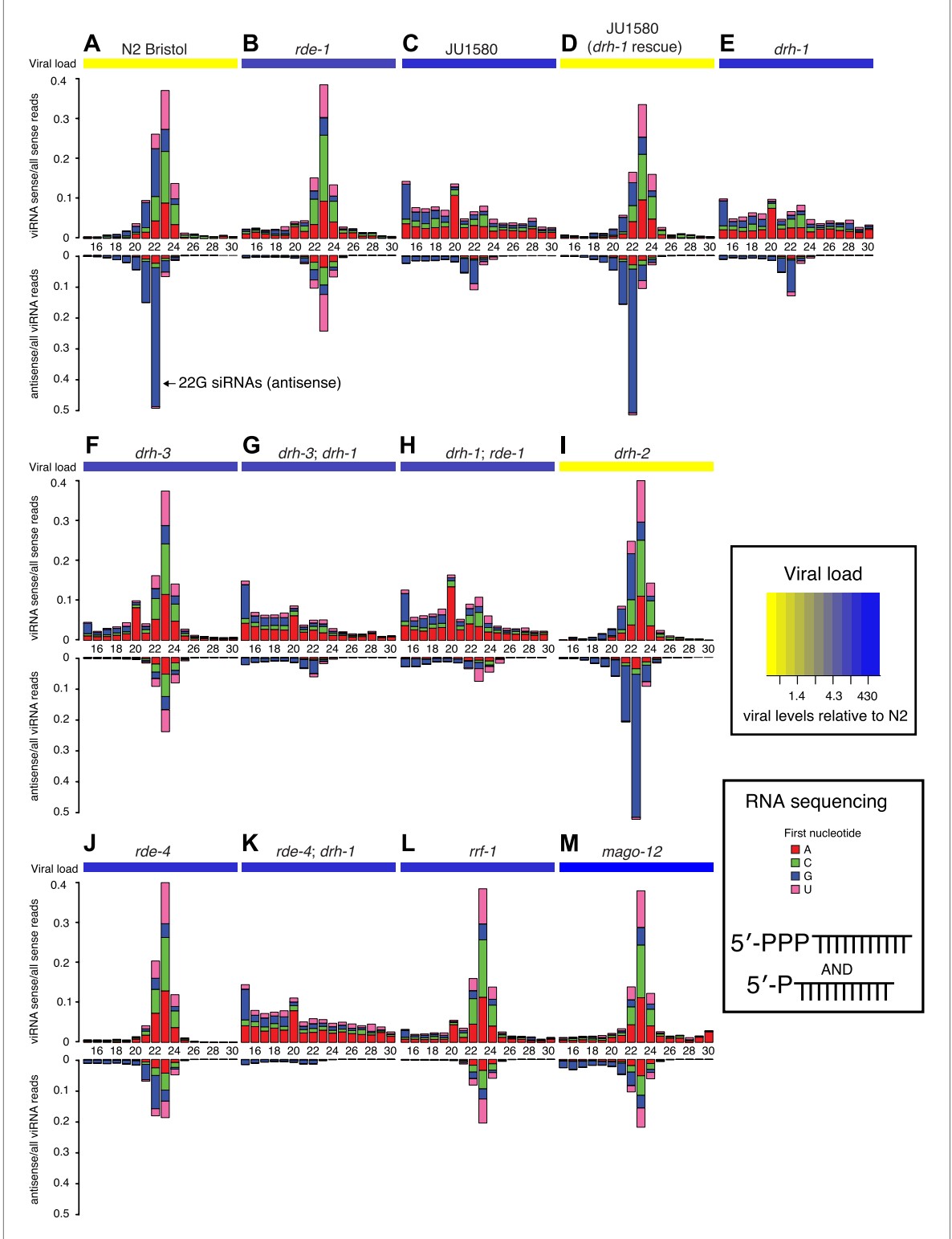

Figure 4. DRH-1 acts upstream of a 22G secondary siRNA pathway. (A–M) Primary and secondary viRNA populations in strains as indicated. 5' independent small RNA sequencing captures 5' primary siRNAs (5' monophosphate) and secondary siRNAs (5' triphosphate). Data are grouped as sense or antisense and according to length and the identity of the first nucleotide. From the same samples viral load was measured by RT-qPCR of the Orsay virus RNA1 genome after 4 days of infection (heatmap, see also *Figure 3A* and *Figure 3—figure supplement 1*).

*Figure 4. Continued on next page*

*Figure 4. Continued*

The following figure supplements are available for figure 4:

**Figure supplement 1**. Distribution of viRNAs along the Orsay genome.

**Figure supplement 2**. Analysis of residual Dicer products in JU1580 mutants.

**Figure supplement 3**. Analysis of 22G-RNAs mapping to endogenous loci.

**Figure supplement 4**. *drh-1* mutants are not hypersensitive to RNAi.

---

concert with DCR-1. The *rde-4; drh-1* double mutant showed a further reduction in primary 23 nucleotide long Dicer products compared to the *drh-1* single mutant (***Figure 3—figure supplement 2***). Furthermore, residual antisense 22G siRNAs in *drh-1* mutants (***Figure 4E***) were mostly absent in *rde-4; drh-1* double mutants (***Figure 4K***). These data support the conclusion that residual DCR-1 activity on viral dsRNA in *drh-1* mutants is dependent on RDE-4. Thus, the double mutant data confirms the position of DRH-1 as an upstream factor essential for the generation of robust levels of antiviral siRNA in response to infection.

## Discussion

Overall our data support a key and unique role for the *C. elegans* RIG-I-like protein DRH-1 in primary siRNA synthesis by either guiding DCR-1 activity to the viral genome or assisting DCR-1 processing of the double-stranded viral RNA. We suggest that the physical interaction between DRH-1 and DCR-1, and the potential for DRH-1 to recognize the viral genome as foreign, possibly through its well-conserved RIG-I domain, may enable DRH-1 to recruit DCR-1 to the double-stranded replicating viral genome and instigate a hierarchical antiviral siRNA response (***Figure 5***). Our data also show that the role of DRH-1 in viral recognition is distinct from that of its paralogs. Given the sequence and domain similarities between DRH-1, DRH-2 and DRH-3, it will be of interest to determine the mode of RNA recognition by DRH-2 and DRH-3 in the future.

Our findings imply that parallel RNAi pathways are involved in recognition of viral infection, aberrant endogenous transcripts and exogenous RNAi (exo-RNAi) in an experimental setting. It is interesting therefore that DRH-1 is found in a protein complex with DCR-1 and RDE-4 even in the absence of infection (***Thivierge et al., 2012***). This might suggest that this complex limits the availability of DCR-1 for exo-RNAi. However, *drh-1* and N2 worms respond equally to exo-RNAi, suggesting that this is not the case (***Figure 4—figure supplement 4***). The constitutive nature of the DCR/DRH-1 complex may allow cells to respond much more rapidly to the presence of viral replication intermediates. Recent observations from Rui Lu and colleagues are in agreement with our findings (***Guo et al., 2013***).

Antiviral small RNAs generated by Dicer are an evolutionarily conserved mechanism for fighting infection by positive-strand RNA viruses (***Aliyari and Ding, 2009***). However, there must be some mechanism to distinguish between the viral genome and cellular RNAs. *Drosophila* and plants have a dedicated Dicer enzyme responsible for viral dsRNA recognition. We suggest that in *C. elegans* DRH-1 may perform this function. The absence of a role for *drh-1* in the endogenous small RNA pathway (***Figure 4—figure supplement 3***; ***Gu et al., 2009***) supports the idea that it encodes a viral-specific recognition factor. In mammals, the C-terminal domain of the DRH-1 ortholog RIG-I is able to recognize the 5′ triphosphate on viral genomes in the context of the double-stranded replication intermediate (***Hornung et al., 2006***; ***Pichlmair et al., 2006***; ***Rehwinkel et al., 2010***). The C-terminal domain is conserved in DRH-1, thus suggesting that DRH-1 might use a recognition mechanism analogous to RIG-I to recruit DCR-1 to the Orsay virus RNA. In support of this, 5′ RACE experiments from total RNA from infected animals using Orsay virus specific primers fail to detect product unless the 5′ triphosphate is removed prior to adaptor ligation (***Figure 4—figure supplement 3***), suggesting that the majority of Orsay genomic RNA molecules are indeed phosphorylated at the 5′ end.

It is interesting that the recognition function of DRH-1 may be conserved with mammalian RIG-I whilst the effector pathways are apparently distinct. It will therefore be intriguing to examine whether any part of the function of DRH-1 in antiviral RNAi might be conserved in mammals. As yet, although small RNA responses have been analyzed in mammalian cells infected with viruses (***Parameswaran et al., 2010***) it is not clear whether these have a significant role in the defense of viral infection or

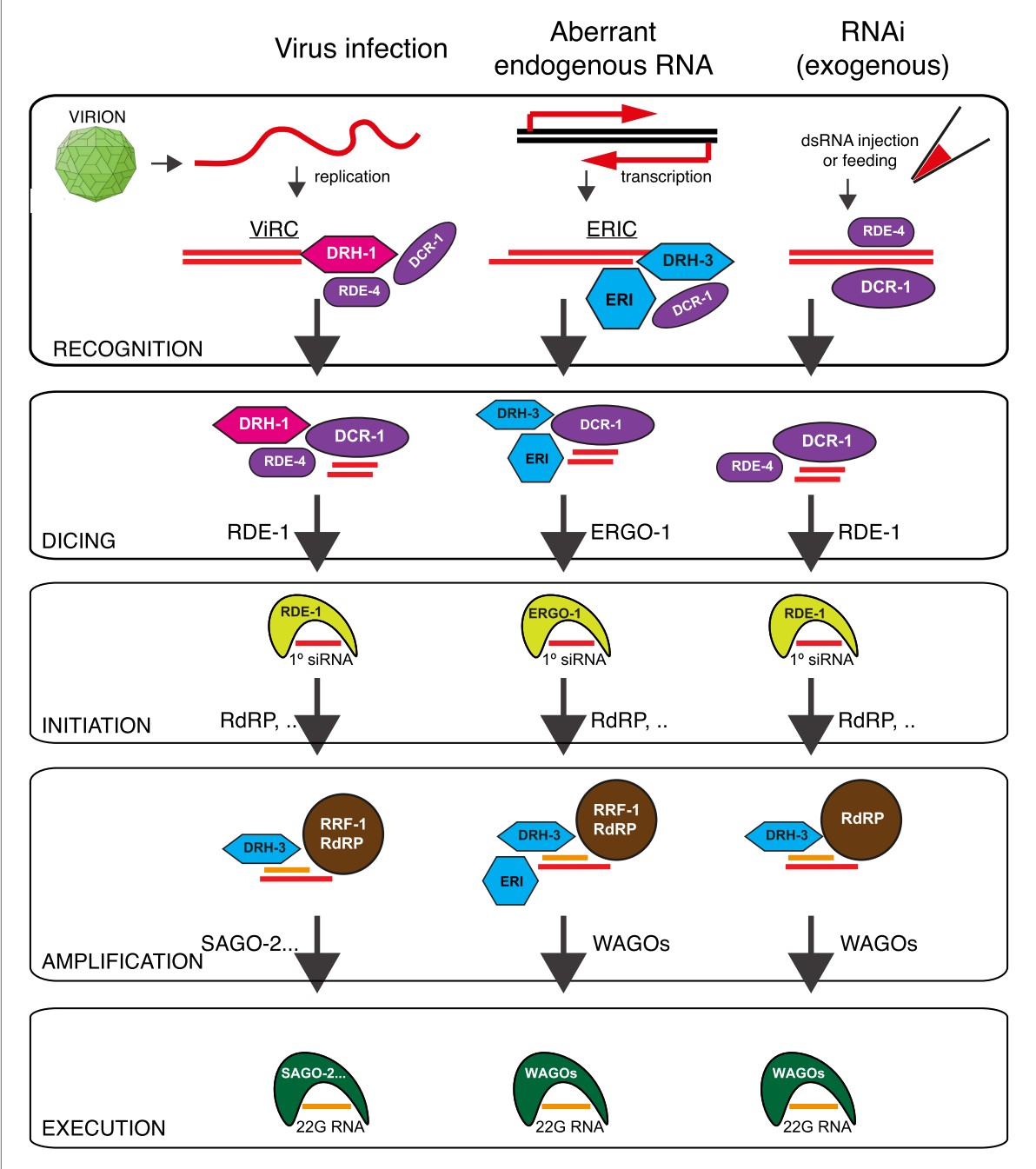

**Figure 5**. Model: DRH-1 triggers a hierarchical antiviral RNAi pathway. Upon infection of the N2 *C. elegans* strain by the Orsay virus, DRH-1 recruits DCR-1 and its partner RDE-4 to the viral dsRNA replication intermediate. DCR-1 cleaves the viral genome into 23 nt viRNA duplexes with a 2 nt 3' overhang. Duplex viRNAs are incorporated into the Argonaute protein RDE-1 and one strand is lost to give rise to primary viral siRNAs (primary viRNAs). Primary viRNAs and RDE-1 recruit an RdRP complex to the viral genome to synthesize secondary viral siRNAs, which act to silence viral transcripts or inhibit virus replication. The antiviral RNAi pathway is dependent on the SAGO-2 secondary Argonaute protein (*Figure 3—figure supplement 1C*). The antiviral RNAi pathway has parallels to the exogenous RNAi pathway and the endogenous RNAi pathway thought to recognize aberrant endogenous transcripts (*Gu et al., 2009*). A complex of DRH-1, DCR-1 and RDE-4 has previously been observed in whole animal lysates (*Tabara et al., 2002*; *Duchaine et al., 2006*; *Thivierge et al., 2012*). We refer to this complex as the Viral Recognition Complex (ViRC). ERI, other ERI factors. ERIC, ERI Complex.

whether they display any similarities to antiviral RNAi pathway in plants or nematodes. Examining whether cells deficient in RIG-I show alterations in small RNA pathways upon viral infection may help to address this question. Equally, it will be interesting to identify whether DRH-1 has a signaling role in *C. elegans* upon viral infection. Such studies might indicate whether the gene expression or the RNAi function is the ancestral role of RIG-I-like genes.

RIG-I and MDA-5 respond to viral infection with changes in cytoplasmic localization that result in the activation of the interferon response (*Nakhaei et al., 2009*). In addition, *RIG-I* and *MDA-5* are themselves interferon-induced genes. It will therefore be of great interest to explore the behavior of DRH-1 upon viral infection. A recent genome-wide analysis of gene expression upon Orsay virus infection in *C. elegans* did not detect any significant changes in *drh-1* transcript levels (*Sarkies et al., 2013*). Furthermore, a GFP-DRH-1 fusion protein did not show marked alterations in expression levels or subcellular localization upon infection with the Orsay virus (our unpublished observations). It will be important to analyze the behavior of the endogenous DRH-1 protein upon infection in future studies.

Our discovery of an inactivating deletion in *drh-1* carried by many wild isolates of *C. elegans* is consistent with the rapid evolution under strong selection known to characterize proteins involved in immunity, including antiviral RNAi defense (*Obbard et al., 2006*; *Vasseur et al., 2011*). Yet it is puzzling that a derived allele with deleterious consequences in the presence of viral infection is found at intermediate frequency. One possible explanation might be the low natural occurrence of infecting viruses, meaning that the *drh-1* deletion is effectively neutral. In support of this argument we have not been able to detect similar intestinal symptoms of viral infection in our extensive *C. elegans* sampling (*Félix and Duveau, 2012*), including in other years of sampling in the Orsay orchard. Alternatively the *drh-1* deletion may have hitch-hiked to fixation with a closely linked beneficial mutation. Such a phenomenon is likely to be common in *C. elegans* natural populations due to the low effective outcrossing rate, which results in high linkage disequilibrium, especially when associated with a positive selective sweep (*Cutter et al., 2009*). Indeed we find that the natural *drh-1* deletion allele is in high linkage disequilibrium with the surrounding region of chromosome IV (*Figure 2E*). A related possibility is that the *drh-1* deletion itself might have a positive fitness effect in some natural conditions. Although we did not detect a fitness advantage for the region surrounding the *drh-1* deletion under laboratory conditions, we cannot test all possible natural conditions, thus a beneficial effect for the *drh-1* deletion or its surrounding region remains a possibility. Interestingly, RIG-I appears to have been lost in several clades, including chickens (*Zou et al., 2009*; *Barber et al., 2010*; *Table 1*), which might explain the increased sensitivity of chickens to avian influenza virus when compared to ducks (*Barber et al., 2010*).

In conclusion, we found that the RIG-I domain has an ancient role in antiviral immunity outside of mammals, yet that this broad conservation in animals is compatible with recurrent losses in insects, ducks, and down to some *C. elegans* isolates. Our results further indicate that the conserved biochemical activity of RIG-I is viral recognition, whereas downstream effector pathways, such as RNAi and interferon responses, may differ between *C. elegans* and mammals.

## Materials and methods

### Genetics
*C. elegans* were grown under standard conditions at 20°C unless otherwise indicated. The wild-type strain was var. Bristol N2 (*Brenner, 1974*). All strains used are listed in *Supplementary file 1A*.

### Virus filtrate preparation
Virus filtrate was prepared as described previously (*Félix et al., 2011*).

### Genome-wide association mapping in wild isolates

#### Choice and infection of the 97 natural isolates of *C. elegans*
We used 97 natural isolates that have been partly sequenced in a previous study using RAD-sequencing (*Andersen et al., 2012*). We first submitted each isolate to a bleaching treatment that eliminates horizontally transmitted symbionts such as the Orsay virus (*Félix et al., 2011*). For each isolate, we infected two 55-mm plates containing five young adults, in triplicate. Cultures were incubated with the Orsay virus at 23°C for 7 days. Maintenance over more than 4 days after infection was performed by transferring a piece of agar (approximately 0.1 cm$^3$) every 2–3 days to a new plate with food. For each infection batch, JU1580 was infected as a control. At 7 days post-infection, nematodes from two plates were collected in M9 and RNA was extracted as described previously (*Félix et al., 2011*).

**Table 1.** Evolution of Dicer and RIG-I family proteins

| | | | | helicase + RIG-I structures in species | Dicer |
|---|---|---|---|---|---|
| Cnidaria | | | *Nematostella vectensis* | 2 | 1 |
| Bilateria | Protostomia | Polyzoa | | | |
| | | Platyzoa | *Schmidtea mediterranea* | 0 | 1 |
| | | Kryptochozoa | | | |
| | | Mollusca | *Aplysia californica* | 0 | 1 |
| | | Annelida | *Platynereis dumerilii* | 0 | 1 |
| | | Ecdysozoa | *Drosophila melanogaster* | 0 | 2 |
| | | | *Trichinella spiralis* | 2 | 1 |
| | | | *Caenorhabditis elegans* | 3 | 1 |
| | Deuterostomia | | *Branchiostoma floridae* | 2 | 1 |
| | | | *Meleagris gallopavo* | 3 | 1 |
| | | | *Taeniopygia guttata* | 3 | 1 |
| | | | *Gallus gallus* | 2 | 1 |
| | | | *Homo sapiens* | 3 | 1 |

Presence of Dicer and RIG-I family proteins in selected animals. Data were obtained from Pfam (version 26.0) (**Finn et al., 2010**) (pfam.sanger.ac.uk). RIG-I family proteins were identified by having both helicase domains and the RIG-I C-terminal domain (Pfam: PF11648). Available sequence data is sparse for some clades and absence of data might not be sufficient evidence for absence of genes.

## RT-qPCR for GWAS
cDNA was generated from 1 µg total RNA with random primers using Superscript III (Life Technologies, Foster City, CA). cDNA was diluted to 1:10 for RT-qPCR analysis. RT-qPCR was performed using LightCycler 480 SYBR Green I Master (Roche, Mannheim, Germany). The amplification was performed on a LightCycler 480 Real Time PCR System (Roche). Each sample was normalized to *eft-2*, and then viral RNA1 (primers GW194 and GW195 in the 97) or RNA2 (oTB17 and oTB18 in the recombinant analysis) levels were compared to the level present in a reference RNA extract obtained from infected JU1580 animals.

### Association mapping
Association mapping was performed using the EMMA package with the default kinship matrix (**Kang et al., 2008**). The mapped trait was the log-transformed mean value of the qPCR on the RNA1 of the Orsay virus.

## Mapping using recombinant F2-derived families
### Production and infection of the F2-derived families
JU1580 hermaphrodites were crossed to N2 males. 11 heterozygous larvae (stage L4) of the F1 cross progeny were picked singly in 35-mm plates and allowed to self. 110 recombinant F2 larvae (stage L4) were picked as individuals into 35-mm Petri dishes. F2 animals were cultured at 20°C for 24 hr and then inoculated with 10 µl of Orsay virus filtrate. After 4 days of culture at 20°C, the F2 animals and their F3/F4 progeny were resuspended into M9 and washed three times in 1 ml of M9. Animals were pelleted in approximately 20 µl of M9 after the last wash. 5 N2 and 5 JU1580 animals were treated in the same way than the recombinant F2 animals to serve as controls.

### RT-qPCR of F2 families
To measure the viral load in the infected F2 families, 5 µl of nematode pellets were added to 45 µl of lysis solution (with 1:100 DNase I) from the Power SYBR Green Cells-to-Ct kit (Ambion, Austin, TX). The lysis mixtures were freeze-thawed 10 times using liquid nitrogen and a hot water bath, and then vortexed for 30 min in 96-well plates. 5 µl of Stop Solution (Ambion) were added to each lysis mixture to

complete the RNA extraction step. The cDNA synthesis and the qPCRs were then performed using the Power SYBR Green Cells-to-Ct kit (Ambion). We used 2 µl of cDNA (equivalent of cDNA contained in 0.09 µl of nematode pellet) as a template for the qPCR. The level of viral RNA was normalized to *gapdh*.

## Sequencing

DNA from JU1580 or pools of 20 N2 × JU1580 recombinant lines was isolated and DNA libraries were prepared for high-throughput sequencing using the Nextera DNA sample prep kit from Illumina. Sequencing was 150 paired end sequencing was performed on a Illumina MiSeq instrument. The JU1580 genomic reads are accessible at the NCBI Short Read Archive under the accession number: SRS369862.

## Sequencing analysis

Processing of Illumina reads, mapping and variation detection was performed using the CLC Genomics Workbench 5 (version 5.5.1). N2 reference genome and annotation used was WormBase release WS220 (www.wormbase.org). SNP and indel analysis was performed using the R statistical environment.

## Introgression of the candidate region into the N2 genomic background

### Production and infection of the recombinant animals

N2 males were crossed to JU1580 hermaphrodites, and the F1 males outcrossed to N2 hermaphrodites. As *niDf250* had already been identified in the genome sequence analysis, F2 animals were genotyped by PCR for the presence of its JU1580 allele in the 6 Mb central region of chromosome IV with primers oTB40 and oTB43 (*Supplementary file 1B*). From one heterozygous animal, 20 homozygous *niDf250* F3 animals were selected and crossed to N2 males to start another cycle of introgression. We repeated this cycle twice and in the final cycle 20 homozygous *niDf250* F3 animals were isolated. We used a pyrosequencing genotyping method (PyroMark Q96 ID instrument; Biotage) to select among these 20 animals those with the full 6 Mb central region of chromosome IV from JU1580 into the N2 genetic background, and only this region. We genotyped six SNPs on chromosome IV (one at either end of the chromosome, one at either end of the 6 Mb region and two within this region), three SNPs on chromosomes I, III, V, X and two on II. We thus obtained an introgressed line, called JU2170. To obtain recombinants within the 6 Mb central region of chromosome IV for fine mapping, JU2170 hermaphrodites were crossed to N2 males. From 20 F1 heterozygous hermaphrodites, we isolated 300 F2 individuals and selected for recombinants between the two SNPs at either end of the region (IV: 3,877,431 and IV: 11,083,410). We thus obtained five recombinants. We further narrowed down the recombination break point by genotyping inside the candidate region (*Figure 1—figure supplement 2*).

### Pyrosequencing

SNPs were genotyped by pyrosequencing, using a PyroMark Q96 ID instrument from Biotage (Uppsala, Sweden). The PSQ Assay Design software was used to design pyrosequencing primers. To biotinylate one strand in the PCR reaction, we added a universal tag on one primer, as described (*Aydin et al., 2006*). For each strain, 10 adult animals were mixed with 10 µl of worm lysis buffer (*Fay and Bender, 2008*; *Félix et al., 2011*) containing proteinase K at 100 µg/ml. After lysis at 60°C for 1 hr and proteinase K inactivation at 95°C for 15 min, we added 1 µl of worm lysate to 50 µl of PCR mix composed of: 0.25 µl of GoTaq DNA polymerase (Promega, Madison, Wisconsin), 5 µl of dNTP at 2 mM, 10 µl of 5x GoTaq buffer, 0.17 µl of non-biotinylated primer at 10 mM, and 0.87 µl of corresponding universal biotinylated primers at 10 mM, and 1 µl of the reverse primer. Single-stranded DNA was then purified and the pyrosequencing reaction was performed following the manufacturer's indications. Primers are listed in *Supplementary file 1B*.

## *drh-1* expression

Non-synchronized animals were cultured in a 55-mm plate and collected in M9 just before starvation. RNA extraction and RT-qPCR were performed as described previously (*Félix et al., 2011*).

## *drh-1* rescue in the JU1580 strain

JU1580 animals were transformed as described (*Mello and Fire, 1995*) with the fosmid WRM0640dC01 that contains the entire length of the *drh-1* gene and its operon CEOP4647. The injection mix contained

10 ng/µl of fosmid DNA, 5 ng/µl of the co-marker transgene *myo-3::gfp::unc-54,* 85 ng/µl of 1 kb DNA ladder (Invitrogen), 20 mM potassium phosphate pH 7.5, and 3 mM potassium citrate pH 7.5. The transgene was then integrated via X-ray irradiation as described (*Fire, 1986*). We controlled that the transgenic copy of *drh-1* was transcriptionally active in transformed animals by qRT-PCR on a portion of the mRNA that is deleted in the wild JU1580 strain.

## Viral load in mutants in the RNAi pathway and in *drh-1* rescued JU1580

### Infection of strains of interest
For all strains mentioned in *Figures 3 and 4*, as well as for the strain PD8753 that carries a balanced mutation in *dcr-1,* one or two young adults were inoculated with 20 µl of viral filtrate for 4 days at 20°C (or 15°C for *mago-12* animals) in 55-mm plates in five biological replicates.

### RT-qPCR on mutant strains
4 days after infection, all animals except those from the PD8753 strain were collected in M9; RNA extraction and RT-qPCR were performed as described previously (2). Aliquots of RNA were kept apart for small RNA libraries (see below). For the infected progeny of PD8753 parents, 16 *dcr-1* and 16 *dcr-1/+* adults per replicate were selected under the fluorescent microscope (i.e., the balancer chromosome carries a *gfp* reporter) and lysed individually in 10 µl of lysis solution (with 1:100 DNase I) from the Power SYBR Green Cells-to-Ct kit (Ambion). The lysis mixtures were pooled for each replicates and the qRT-PCR was performed as indicated above for the F2 families.

## Small RNA sequencing

### Library preparation and sequencing
Preparation of RNA used for small RNA libraries is described above. For 5′ independent libraries, 3–5 µg of total RNA was pre-treated with 5′ polyphosphatase (Epicenter) following the manufacturer's instructions. Small RNA libraries were generated from either polyphosphatase treated or total RNA (3–5 µg) using the TruSeq Small RNA kit (Illumina) following the manufacturer's instructions. Small RNA libraries were sequenced using the Illumina MiSeq. Small RNA sequence data were submitted to the Gene Expression Omnibus (GEO) under accession number GSE41693.

### Sequencing analysis
Small RNA libraries were sequenced using the Illumina MiSeq. Fastq files generated by the machine had adaptors removed using the program Cutadapt v1, and were converted into Fasta files using a custom Perl script. For alignment to the viral genome, Fasta files were trimmed to leave only reads of between 15 and 30 nucleotides using a custom Perl script and were aligned using Bowtie (version 0.12.7) to RNA1 and RNA2 of the Orsay virus genome (*Félix et al., 2011*), reporting only the best single alignment with up to one mismatch allowed. Sam files from Bowtie were converted into bam files using the SAMtools utility (*Li and Durbin, 2009*) and bam files were converted into bed files using the BedTools utility (*Quinlan and Hall, 2010*). Bed files were read into the R environment and plots of the length and first nucleotide as well as read distribution along RNA2 were generated using custom scripts written in R. The level of virus as measured by qRT-PCR was observed to correlate non-linearly with the total level of sense small RNAs in different N2 wild-type samples, thus data were normalized to all viral siRNAs or all sense viRNAs to illustrate differences in read distribution between different samples. Analysis of the overlap to identify potential DCR-1 signatures was carried out by selecting reads of the same length coming from opposite strands on the viral genome, which overlapped by at least one nucleotide. The number of reads overlapping by every possible length either 3′ or 5′ was compared to that expected were the entire distribution uniform using a $\chi^2$ test. Only one overlap was statistically significantly enriched with a cut-off of $p<0.05$ as indicated in the text. For alignment to the *C. elegans* genome to analyze any potential changes in 22G levels, fasta files were trimmed to leave only reads of between 18 and 30 nucleotides, and the fasta files were collapsed using the FastX Toolkit. Reads matching to microRNA precursors downloaded from miRbase (*Kozomara and Griffiths-Jones, 2011*) were removed using a custom Perl script, and the remaining reads were aligned to the Ce6 genome (WS190) using Bowtie, reporting the best match with no mismatches (parameters—best–k 1–v 0). After converting bam files to bed files as above, 22Gs mapping antisense to the Ce6/WS190 UCSC annotations of genes (sangerGene.txt) downloaded from the

UCSC genome browser website, were then selected from the alignments using a custom Perl script. Genes with abundant 22G reads in N2 wild-type worms were further selected by setting a cut-off of at least five reads in at least one of the N2 wild-type samples and duplicate or overlapping annotations were removed. Comparison of mutants to N2 wild-type was carried out after normalizing to the total number of aligned reads in each library. Secondary structure analysis of the viral genome was performed using RNA-fold (*McCaskill, 1990*) on a 50 bp window sliding by increments of 20 bp. A Z-score for the secondary structure strength of each window was calculated by comparing the ensemble free energy mean ensemble free energy for 100 random shuffles of the 50 bp window.

## 5′ RACE analysis of the Orsay virus

Total RNA isolated from infected animals as described above was ligated to the 5′ adaptor from the Illumina Truseq small RNA kit either with or without prior treatment with 5′ polyphosphatase. Only monophosphorylated 5′ ends will be able to ligate to the adaptor, as for the small RNA sequencing. Ligated products were then reverse transcribed using a primer specific for RNA1. The resulting cDNA was then analysed by standard Taq PCR and gel electrophoresis using either primers designed to amplify within the 5′ end of the viral genome as a positive control or primers to amplify from the adaptor into the 5′ end, thus determining whether the adaptor was able to ligate efficiently to the 5′ end (*Figure 4—figure supplement 3B*).

## Molecular evolution analysis of the 6 Mb central region of chromosome IV

For each of the 97 isolates, 97 SNPs included in the 6 Mb central region of chromosome IV associated with Orsay virus sensitivity were extracted from RAD-sequencing data (*Andersen et al., 2012*). DNAsp allowed us to classify all isolates in 28 different haplotypes for this region. The neighbor-net network (*Bryant and Moulton, 2004*) was then drawn using the SplitTree software (*Librado and Rozas, 2009*). The average number of differences per polymorphic RAD site along chromosome IV was calculated using DNAsp (*Huson and Bryant, 2006*) using a sliding window of 25 SNPs every 10 SNPs. The linkage disequilibrium (*D′*) between *drh-1* alleles and polymorphic RAD sites on chromosome IV was calculated by DNAsp.

## Progeny and longevity assays

### Progeny production

For each strain, we seeded ten 55-mm NGM plates with five L4 animals on each. 5 of the 10 plates were infected with the Orsay virus. Before starvation was reached, we transferred a piece of agar (4 × 4 mm at the surface, 1 cm to the bottom of the plate) from each of the five plates of each treatment (to maximize infection probability) to a single 90 mm NGM plate. 40 L4 animals (F2 progeny from the infected animals) were then isolated for each treatment. Each scored individual was transferred into a new plate 24, 36, 48, 60, 72, 96 and 120 hr after the L4 stage, and progeny number was scored 48 hr after each transfer (most were then at the L4 and adult animals). To ease scoring, some plates were cooled to 4°C after 48 hr and scored within 2 days. N2 and the *drh-1* mutant in the N2 background (RB2519) were assayed in parallel. JU1580 and JU1580 rescue (SX2377) were independently blind tested in parallel.

### Longevity assay

From the same starting populations, we in addition isolated for the survival assay ten animals on nine 55 mm diameter NGM plates (n = 90). Each pool of 10 animals was then transferred to a new plate every 24 hr. Survival was assessed every day by scoring the capacity of each animal to react to stimulation (shaking plates and touching them with a pick). Dead animals were removed from the plate. We also measured the lifespan of each parent from the progeny scoring (n = 40).

### Infection rates

Each F2 population was fixed and prepared for FISH as previously described (*Kowalinski et al., 2011*) using custom Stellaris (Biosearch Technologies) probes for the Orsay virus RNA1 molecule (*Supplementary file 1B*) labeled with Quasar 670 Dye. Standard fluorescence microscopy was performed using an upright Zeiss AxioImager M1 equipped with a Pixis 1024B camera (Princeton instruments) and a Lumen 200 metal arc lamp (Prior Scientific). A L4 larva was scored as infected when we could see a signal inside at least one intestinal cell. The proportion of infected L4 individuals was the following: 35.0% (n = 109) of JU1580; 3.4% (n = 121) of SX2377; 53.0% (n = 103) of RB2519; 9.0% (n = 100) of N2 animals.

## Statistical analysis

The lmer function from the R package « lme4 » was used to determine the effect of the treatment on the dynamics of progeny production by comparing with an ANOVA the Akaike Information Criterion of a model involving both time, treatment and their interaction to that of a null model involving only time. For the longevity assay, we used the survdiff function (log rank test) of the R package « survival » as to test the effect of the treatment.

## Competition assay

### Competition

The N2 and JU2196 genotypes were competed in the presence or absence of the virus. JU2196 is a near isogenic line with the 4.4 Mb central part of chromosome IV from JU1580 into the N2 background (*Figure 1—figure supplement 2A*). Both strains were cleaned and roughly synchronized by bleaching. Once the bleached embryos had developed to adulthood, we let them lay embryos for 1 hr and half to further synchronize the populations. After letting the embryos hatch overnight, we started the competition experiment with L2 hermaphrodite larvae in a 1:1 ratio (20 plates with 10 N2 and 10 JU2196 animals on each), adding 50 µl of Orsay virus preparation on half of these plates and 50 µl of M9 solution in the other. Populations were harvested 5 days later with M9 solution under sterile conditions in microfuge tubes. After 2 min of centrifugation at 3000 rpm, the supernatant was removed and a fraction of the pellet containing 100–400 animals (generally 2 µl) was transferred to a new NGM plate. After this initial transfer, transfers were repeated approximately every 36 hr, before starvation. We did not observe a high occurrence of males in this experiment, thus it is unlikely that the tested chromosome IV region recombined.

### Genotyping

At each transfer, 2 µl of the pellet was mixed with 18 µl of worm lysis buffer containing proteinase K at 100 µg/ml. The product of the worm lysis was then used as a PCR template using the pyrosequencing primers IV_6124501 Forward and Reverse (*Supplementary file 1B*). We quantified the proportion of the *drh-1*(N2) allele using the quantitative option of the pyrosequencer (PyroMark Q96 ID instrument; Biotage). To measure the accuracy of this quantification method, a standard curve was performed with different known proportions of alleles (known proportions of genomic DNA). The correlation between observed and real allele frequencies exceeded 0.983 and the average standard deviation calculated from five replicates of observed frequencies was 6%.

### Statistical analysis

The lmer function from the R package « lme4 » was used to determine the effect of the treatment on the genotype frequency by comparing with an ANOVA the Akaike Information Criterion of a model involving both time, treatment and their interaction to that of a null model involving only time.

## RNAi

*unc-22* and empty vector RNAi bacteria were grown for 6 hr at 37°C. *unc-22* bacteria were then serially diluted with the empty vector bacteria and seeded onto NGM agar plates containing IPTG (1 mM) and carbenicillin (25 µg/ml). After drying overnight, N2, *drh-1* or *eri-1* worms were added and then grown at 20°C for 4 days. Each strain was tested in triplicate at each dilution and 15 animals selected at random from each plate were scored for either twitching or paralysis phenotypes.

## Acknowledgements

We thank M Tanguy and J Rehwinkel for critical insights into viral biology, C Bradshaw and G Allen for Bioinformatics support and S Moss, I Nuez and A Richaud for laboratory management. We are grateful to C Franz and the D Wang laboratory for viral filtrate preparations, to E Andersen for help with the whole-genome association, and to M De Bono for reagents. Some strains were provided by the CGC, which is funded by NIH Office of Research Infrastructure Programs (P40 OD010440). Some deletion mutations were provided by the International *C. elegans* Gene Knockout Consortium which is funded by the National Institute of Health, the Canadian Institute for Health Research, Genome Canada, Genome BC, and the Michael Smith Foundation. Fosmids were provided by D Moerman laboratory. AA was supported by a Herchel-Smith postdoctoral fellowship. PS was supported by a research fellowship at Gonville and Caius College, Cambridge.

## Additional information

### Funding

| Funder | Grant reference number | Author |
|---|---|---|
| Wellcome Trust | 092096 | Jérémie Le Pen, Eric A Miska |
| Cancer Research UK | RG57329 | Eric A Miska |
| European Research Council | RG58558 | Eric A Miska |
| Centre National de la Recherche Scientifique | | Marie-Anne Félix, Tony Bélicard |
| Agence Nationale pour la Recherche | ANR 11 BSV3 01301 | Marie-Anne Félix, Lise Frézal |
| Coup d'Elan de la Fondation Bettencourt-Schueller | | Marie-Anne Félix |
| Ecole Normale Supérieure | | Marie-Anne Félix |

The funders had no role in study design, data collection and interpretation, or the decision to submit the work for publication.

### Author contributions

AA, TB, JLP, PS, Conception and design, Acquisition of data, Analysis and interpretation of data, Drafting or revising the article; LF, Acquisition of data, Analysis and interpretation of data, Drafting or revising the article; NJL, Acquisition of data, Contributed unpublished essential data or reagents; M-AF, EAM, Conception and design, Analysis and interpretation of data, Drafting or revising the article

## Additional files

### Supplementary files

• Supplementary file 1. (**A**) Table 1: Strains used in this study. (**B**) Table 2: Oligonucleotides used in this study.

### Major datasets

**The following datasets were generated:**

| Author(s) | Year | Dataset title | Dataset ID and/or URL | Database, license, and accessibility information |
|---|---|---|---|---|
| Sarkies P, Miska EA | 2013 | Analysis of small RNA response to viral infection in *C. elegans* | GSE41693; http://www.ncbi.nlm.nih.gov/geo/query/acc.cgi?acc=GSE41693 | Publicly available at GEO (http://www.ncbi.nlm.nih.gov/geo/). |
| Sarkies P, Miska EA | 2013 | Sequencing of a *C. elegans* wild isolate JU1580 | SAMN01766984; http://www.ncbi.nlm.nih.gov/biosample/?term=SAMN01766984 | Publicly available at the NCBI BioSample database (http://www.ncbi.nlm.nih.gov/biosample). |

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
