## [Decision Letter]

Thank you for sending your work entitled “A deletion polymorphism in the
*C. elegans* RIG-I homolog disables viral RNA dicing and
antiviral immunity” for consideration at *eLife*. Your article
has been favorably evaluated by a Senior editor (Detlef Weigel) and 2 reviewers.

The Senior editor has assembled the following comments to help you prepare a revised
submission.

Ashe and colleagues use an appealing mix of quantitative and molecular genetics to
identify and characterize a natural polymorphism in the *C. elegans*
DRH-1 gene that affects production of viral-derived primary siRNAs. The gene emerged
from an analysis of natural variation in sensitivity to the Orsay virus, and by a
combination of GWAS, introgression mapping, and whole-genome sequencing, the authors
managed to pin down a small deletion as a major-effect locus. Using mostly if not
exclusively genetics, the authors provide evidence that DRH-1 is likely to
specifically recruit the Dicer protein DCR-1 to viral dsRNA substrates. This clearly
is a new finding, especially in an organism with a single Dicer. While we find the
work very interesting, a minimal set of additional experiments would help support
the inferences of the genetic analyses by answering the following questions: 1) Does
viral infection affect the expression and subcellular location of DRH-1 protein, as
assayed with a GFP fusion?

2) Is there evidence that EXO RNAi is enhanced in *drh-1* mutants
because absence of DRH-1-bound Dicer would increase Dicer pools? That DRH-1 is not
involved in endosiRNA production is consistent with a specific role for SRH-3 in
bringing Dicer onto endosiRNA substrates. If the answer to the initial question is
yes, this could support the scenario for the possible benefit of the
*drh-1* deletion in a significant fraction of the wild accessions
because of enhanced environmental RNAi (which may confer unknown fitness
advantages).

3) A critical experiment for supporting the proposed model would be whether the
naturally found truncated form of DRH-1 indeed fails to interact with DCR-1 and
RDE-4. We understand that antibodies are not available to test this. Thus,
mechanistic inferences about DRH-1 action should be toned down in the revision.

---

## [Author Response]

*1) Does viral infection affect the expression and subcellular location of
DRH-1 protein, as assayed with a GFP fusion*?

With regard to changes in *drh-1* mRNA levels we have recently
published a study analysing gene expression changes upon Orsay virus infection
(Sarkies et al., Genome Research 2013). We observed no statistically significant
changes in *drh-1* transcript levels upon infection in wild-type (N2)
animals. Thus *drh-1* is unlikely to be transcriptionally regulated
upon viral infection.

With regards to the subcellular localization of DRH-1 protein, we agree with the
reviewers that the cell biology of viral infection in *C. elegans* is
of great interest, including changes in the subcellular localization of DRH-1 before
and after viral infection. However, we do not agree that this analysis relates
directly to the manuscript that we have submitted: any change or lack of change in
subcellular localization would neither strengthen nor contradict our model.
Furthermore, we believe that a thorough analysis of DRH-1 behaviour upon infection
would have to include analysis of the subcellular dynamics of the Orsay virus in
order to be meaningful. Since we do not fully understand this yet, a combined
analysis should be the focus of an independent future study. Nevertheless, we have
generated a transgenic line expressing a DRH-1-GFP fusion protein. We found that
these transgenic animals localize GFP diffusely in the cytoplasm of intestinal and
other cells (Author response image 1 below, transgene driven by a
“ubiquitous” promoter, *sur-5*). We observed that the
cytoplasmic localization of GFP does not change in response to infection with the
Orsay virus. However, in the absence of a suitable antibody that works in
immunofluorescence, we are unable to assess the localization of the endogenous
protein, thus we cannot rule out that changes in DRH-1 subcellular localization
occur upon infection. We should therefore be cautious about the interpretation of
such a negative result and we would prefer not to include these data in the
manuscript. We have therefore added a paragraph to the Discussion referencing the
gene expression paper, and we have referred to the results of the DRH-1GFP fusion
protein analysis as data not shown.Author response image 1.Assessing the subcellular localization of GFP in
*drh-1* mutant animals carrying a
*sur-5::drh-1::gfp* transgene as an extra-chromosomal
array.Infections were carried out as described in Materials and methods. As the
transgene is extrachromosomal, it is lost in some animals (top animal,
top panel). We do not observe a change in GFP intensity or localization
upon infection with the Orsay virus.

*2) Is there evidence that EXO RNAi is enhanced in* drh-1
*mutants because absence of DRH-1-bound Dicer would increase Dicer pools?
That DRH-1 is not involved in endosiRNA production is consistent with a specific
role for DRH-3 in bringing Dicer onto endosiRNA substrates. If the answer to the
initial question is yes, this could support the scenario for the possible
benefit of the* drh-1 *deletion in a significant fraction of the
wild accessions because of enhanced environmental RNAi (which may confer unknown
fitness advantages)*.

The reviewers have raised an interesting point. We therefore performed RNAi
experiments using a dilution series of *unc-22* RNAi (see Materials
and methods) and found that exo-RNAi is not enhanced in *drh-1*
mutants, while it is enhanced in *eri-1* mutants. We conclude that
there is no evidence that DCR-1 interaction with RDE-4 and DRH-1 limits its
availability for exo-RNAi. We have included this data as Figure 4—figure supplement 4. We also added a
discussion point to the Discussion.

*3) A critical experiment for supporting the proposed model would be whether
the naturally found truncated form of DRH-1 indeed fails to interact with DCR-1
and RDE-4. We understand that antibodies are not available to test this. Thus,
mechanistic inferences about DRH-1 action should be toned down in the
revision*.

The truncated DRH-1 (*nDf250*) might fail to interact with
RDE-4/DRH-1, the Orsay RNA, or both. By analogy to RIG-I we suggest that DRH-1
(*niDf250*) fails to recognize the Orsay RNA. However, we agree
with the reviewers: we have toned down our mechanistic inferences by suggesting that
DRH-1 may be necessary for either recognition of viral RNA or correct Dicer
processing “assisting DCR-1 processing of the double-stranded viral
RNA.”